# Therapeutic Potential of Cannabinoid Profiles Identified in *Cannabis* L. Crops in Peru

**DOI:** 10.3390/biomedicines12020306

**Published:** 2024-01-29

**Authors:** Pedro Wong-Salgado, Fabiano Soares, Jeel Moya-Salazar, José F. Ramírez-Méndez, Marcia M. Moya-Salazar, Alfonso Apesteguía, Americo Castro

**Affiliations:** 1CANNAVITAL, Clínica Especializada en Terapias con Cannabinoides, Lima 15022, Peru; farmacia@cannavital.com.pe; 2RENATU Research Group, Faculty of Pharmacy and Biochemistry, Universidad Nacional Mayor de San Marcos, Lima 15039, Peru; fabiano@reaja.com.br (F.S.); cecannabisperu@upch.pe (A.C.); 3Centro de Estudios del Cannabis del Perú, Lima 15022, Peru; marciamilagros21@gmail.com; 4REAJA Laboratory, Curitiba 80045-180, Brazil; 5Faculties of Health Science, Universidad Privada del Norte, Lima 15001, Peru; 6Cannabis and Stone Unit, Nesh Hubbs, Lima 15001, Peru; 7Centro de Información, Control Toxicológico y Apoyo a la Gestión Ambiental CICOTOX, Faculty of Pharmacy and Biochemistry, Universidad Nacional Mayor de San Marcos, Lima 15039, Peru; japesteguiai@unmsm.edu.pe

**Keywords:** medical cannabis, tetrahydrocannabinol, therapeutic potential, chemical profile, cannabinoids, inflorescences, Peru

## Abstract

Cannabis is a plant that is cultivated worldwide, and its use is internationally regulated, but some countries have been regulating its medicinal, social, and industrial uses. This plant must have arrived in Peru during the Spanish conquest and remains widely cultivated illicitly or informally to this day. However, new regulations are currently being proposed to allow its legal commercialization for medicinal purposes. Cannabis contains specific metabolites known as cannabinoids, some of which have clinically demonstrated therapeutic effects. It is now possible to quantitatively measure the presence of these cannabinoids in dried inflorescences, thus allowing for description of the chemical profile or “chemotype” of cannabinoids in each sample. This study analyzed the chemotypes of eight samples of dried inflorescences from cannabis cultivars in four different regions of Peru, and based on the significant variation in the cannabinoid profiles, we suggest their therapeutic potential. The most important medical areas in which they could be used include the following: they can help manage chronic pain, they have antiemetic, anti-inflammatory, and antipruritic properties, are beneficial in treating duodenal ulcers, can be used in bronchodilators, in muscle relaxants, and in treating refractory epilepsy, have anxiolytic properties, reduce sebum, are effective on Methicillin-resistant Staphylococcus aureus, are proapoptotic in breast cancer, can be used to treat addiction and psychosis, and are effective on MRSA, in controlling psoriasis, and in treating glioblastoma, according to the properties of their concentrations of cannabidiol, cannabigerol, and Δ9-tetrahydrocannabinol, as reviewed in the literature. On the other hand, having obtained concentrations of THC, we were able to suggest the psychotropic capacity of said samples, one of which even fits within the legal category of “non-psychoactive cannabis” according to Peruvian regulations.

## 1. Introduction

With the arrival of the Spanish conquerors in the southern part of the American continent in the 15th century, some important plant species were introduced. Cannabis is considered an introduced plant, and is a shrub with industrial value (in textiles and paper), nutritional value (in seeds), and medicinal value (in inflorescences). Cannabis is considered native to the Euro-Asian region and its use has been documented since the earliest Chinese medical references from approximately 5000 years ago [1].

Cannabis exclusively contains a large number of terpenophenolic compounds called “cannabinoids”, which act as partial agonists to specific receptors within the nervous and immune systems of humans, as well as other mammalian and vertebrate species [2]. Cannabinoids that have demonstrated conclusive clinical effectiveness are Δ9-tetrahydrocannabinol (Δ9-THC), which has analgesic, anti-inflammatory, and antiemetic effects that help with chemotherapy-induced nausea and vomiting and with the management of chronic pain and muscle spasticity caused by multiple sclerosis; and cannabidiol (CBD), which has attenuating effects on refractory epileptic syndromes in children, specifically Dravet and Lennox–Gastaut syndromes [3,4,5].

In the late 20th century, it was discovered that our bodies produce molecules derived from arachidonic acid that function as regulators in the nervous and immune systems, similar to what had been demonstrated with phytocannabinoids. These molecules were named endogenous cannabinoids or “endocannabinoids”, and together with the system of receptors distributed throughout the body and the enzymes that mediate their presence, they were called the “Endocannabinoid System” [6], more recently referred to as the “Endocannabinoidome” at a broader level [7]. This system plays a regulatory role in promoting homeostasis.

An important bibliographic reference that describes the therapeutic effects of the main metabolites found in cannabis is an American Herbal Pharmacopoeia 2014 review entitled “Cannabis Inflorescence” [8]. The information within the review includes nomenclature, constituents, identification, and analysis.

Since Canada passed medicinal regulations in 2001 and then allowed nonmedicinal use in 2018, several countries in the American continent have followed suit [9]. In some countries, regulations perform strict control over THC presence or content, such as in the Brazilian case, where pharmaceutical products derived from cannabis must not exceed 0.2% in weight [10], and the pharmaceutical industry must perform their manufacture. In other countries, regulations promote various pathways of access to patients, even personal cultivation, in addition to allowing pharmaceutical products to be available through conventional channels, as is the case in Argentina [11].

Peru has made significant progress with both approaches, as its regulations allow the use of THC without limits for prescription and artisanal production through patient associations and the pharmaceutical industry. Unfortunately, the application of these regulations is not efficient; after six years of regulation, the Peruvian market continues to rely on foreign raw materials at higher costs, maintaining limited access and availability to less than 30% of registered patients in Peru [12].

### 1.1. Regulation and Cannabis Cultivars in Peru 

The use of this plant was restricted in Peru with the publication of Decree Law No. 22095, the Law for the Repression of Illicit Drug Trafficking, in 1978 [13]. Subsequently, it was completely forbidden by the Supreme Decree DS 023-2001SA (Regulation of Narcotics and Psychotropics) issued by the Ministry of Health in 2001 [14]. However, cannabis is considered the most commonly used illegal “substance” or “drug”, and its use is likely increasing in line with the worldwide trend [15]; antidrug authorities report the lifetime prevalence in the population is 14.8 [16].

Extensive areas of the country with favorable geographical conditions for cannabis cultivation have been identified in the Peruvian highlands (Ancash, Huánuco, Junín, and Cajamarca) and on the coast (Lima and La Libertad). However, there have been no publications of reports about the characterization and identification of nationwide cannabis varieties or cultivars as have been performed in other countries [17,18,19,20]. These cannabis cultivars throughout Peru must be the result of the plant’s acclimatization to different geographical regions. The cultivars of cannabis could have been introduced with the arrival of the Spanish during the time of conquest and could have been maintained by farmers (before the prohibition) and “narcotraffickers” (after the prohibition). However, this process has not been properly documented in Peru. Antidrug authorities have identified two kinds of local demand: (a) for recreational purposes, and (b) for medicinal purposes [21]. 

Therefore, considering the risk of losing relevant information about national cultivars as a result of the implementation of the current medicinal regulations, in which licensed producers will introduce foreign genetics due to these regulations, it is necessary to identify, characterize, and quantitatively analyze the cultivars found within the national territory.

### 1.2. Phytocannabinoids

The predominant phytocannabinoid in the cannabis plant is Δ9-THC, synthesized and elucidated (chemical structure) by Rafael Mechoulam in 1964 [22]. This molecule is the product of the decarboxylation of tetrahydrocannabinolic acid (THCA), a non-psychotropic metabolite that originates from the biosynthetic pathway of cannabigerolic acid (CBGA) by the action of the enzyme THCA synthase [16]. Δ9-THC is known for its psychotropic, analgesic, antiemetic, orexigenic, and anti-inflammatory properties. Δ9-THC is a partial agonist of CB1 receptors, which are predominantly expressed in the central nervous system, and CB2 receptors, which are primarily located in cells with immunological functions. Its pharmacokinetics depend on the route of administration, with differentiation between the hepatic metabolic (oral), sublingual, dermic, and pulmonary routes. In 1985, Dronabinol (synthetic Δ9-THC) was approved by the United States Food and Drug Administration (FDA) for the oral management of chemotherapy-induced nausea and vomiting and anorexia in HIV patients [23].

CBD is another 21-carbon terpenophenolic compound that forms after the decarboxylation of its precursor, cannabidiolic acid (CBDA) [24]. It is one of the most abundant non-psychotropic cannabinoids in the cannabis plant. It acts as an agonist on TRPV1 and 5-HT1A receptors to enhance adenosine receptor signaling [25]. Its neuroprotective, antioxidant, antipsychotic, and anticonvulsant actions are among the most important effects observed. In June 2018, the FDA approved the first drug derived from the cannabis plant, called EPIDIOLEX^®^ (Jazz Pharmaceuticals, Dublin, Ireland), an oral solution predominantly containing CBD [26]. Cannabigerol (CBG) is the nonacidic form of cannabigerolic acid (CBGA), considered a non-psychotropic phytocannabinoid that antagonizes the CB1 and CB2 cannabinoid receptors. It is an agonist of the alpha-2 adrenergic receptor and a moderate antagonist of the 5HT1A receptor. This compound can also antagonize transient receptor potential vanilloid (TRPV) receptors, stimulating TRPV1, TRPV2, TRPA1, TRPV3, and TRPV4 [27]. A list of therapeutic uses is shown in Figure 1.

### 1.3. Cannabis Chemical Profiles or “Chemotypes”

Phytocannabinoids are metabolites exclusively found in cannabis, and some of their therapeutic effects are famous. However, only a few medical areas currently classify these effects as having “conclusive efficacy” in systematic reviews on conditions like chronic pain, nausea and vomiting, muscle spasticity, refractory epileptic syndromes, and anxiety [28], with the main responsible metabolites being Δ9-THC, CBD, and to a lesser extent CBG. Additionally, the presence of certain terpenes such as beta-caryophyllene, myrcene, limonene, terpinolene, alpha-pinene, and alpha-terpineol seems to contribute to the medicinal effects.

There is a wide variety of Cannabis cultivars distributed worldwide, and one of the most useful ways to classify them (especially for medicinal purposes) is by using the chemical profile or “chemotype” of their main cannabinoids and terpenes in the female inflorescences. It is then possible to infer the potential therapeutic effects of cannabis cultivar samples by determining their chemotype through quantitative cannabinoid analysis.

Since cannabis has not yet returned to the pharmacopeias of countries with strict health surveillance, it is necessary to consider herbal pharmacopeias as a reference for health use. One of the most notable is the American Herbal Pharmacopoeia FHA, which published a monograph on cannabis inflorescences in 2014 [8]. This document is also included in the “List of bibliographic references supporting the safety of use and traditional use of natural resources or their associations” of the Ministry of Health of Peru [29].

Reports on Peruvian cannabis cultivars have been conducted solely by the police for eradication purposes, and thus no quantitative chemical analyses have been performed. Therefore, it is important to conduct research that can identify and quantify the existing cultivars in the country.

In this study, we identified cannabis cultivars in four Peruvian regions (three in the highlands and one on the coast). Additionally, we quantified the cannabinoids of the collected samples by developing a gas chromatography technique in order to generate a prediction of therapeutic potential based on the cannabis monograph of FHA and an updated literature review with references of higher relevance in the field, carried out by the author.

## 2. Materials and Methods

### 2.1. Study Design and Settings

We designed an observational study based on the analysis of the cultivars following the recommendations of the STROBE guidelines [30]. Peru is a South American country with approximately 33 million inhabitants and has three natural regions (coast, highlands, and jungle) with extensive diversity in flora and fauna. Geographic data are shown in Table 1. The collection of species and chemotype analysis were carried out with cannabis samples from four regions of Peru (Ayacucho and Cajamarca, located geographically in the highlands), and the study was conducted during the second half of 2021.

### 2.2. Crop and Cannabinoid Identification

#### 2.2.1. Crops and Inclusion Criteria

In each region (Figure 2), inflorescence samples were collected (a total of 8 samples) from adult horticulturists and farmers who conserve “local varieties” and, although these producers allowed the sampling procedure, they refrained from providing extra data due to the risks concerning legal interpretation of cannabis cultivation without commercial purposes (trafficking).

We included Peruvian crops from willing local producers in Peru. These crops comprised “local varieties”, untainted by illicit trafficking or trade, and were accessible for research on cannabis plants throughout the year 2021. Fresh inflorescences were obtained from each cultivar, and in one case, a complete plant was allowed to be collected by using simple traction [31]. Before each sampling, a survey of the cultivation area was conducted to include plants over 1 m in size that were in the flowering stage and ready for harvest. We performed a simple random sampling and included unfertilized female cannabis plants in the flowering stage (without seeds).

#### 2.2.2. Botanical Characterization and Cannabinoid Quantification

The collected inflorescence samples were stored and transported from the sampling zones in each region to the Faculty of Pharmacy and Biochemistry of the Universidad Nacional Mayor de San Marcos (UNMSM) in Lima. Additionally, the only sample of a complete plant was sent to the Museum of Natural History of UNMSM. The samples were identified as *Cannabis* L., and the inflorescences were dried following a previous protocol [32]. The experimental methodology for obtaining standardized extracts is shown in Figure 3. We used 130 mg of dried cannabis inflorescences from each cultivar for chromatographic analysis, following international recommendations [33]. 

The equipment used was a gas chromatograph coupled with a flame ionization detector (FID), model 7890A (Agilent, Santa Clara, CA, USA), using a fused silica capillary rtx-5MS column with dimensions 30 m × 0.25 mm × 0.25 mm and helium as the carrier gas, with an 18 min analysis time [34]. Quantitative chemical analysis was performed for the main secondary metabolites (THC, CBD, CBG, and CBN), for which imported standards were available. Official cannabis sample identification and cannabinoid quantification had never been performed in Peru before this study (Certificate of Registration of the Museum of Natural History Nº012-USM-2020).

#### 2.2.3. Potential Therapeutic Profiles

The chemotypes of cannabinoids found in each sample of cannabis inflorescences were related to the therapeutic effects described for each cannabinoid in the FHA monograph [8]. In this way, potential therapeutic effects were proposed for each of the eight cannabis samples based on the concentrations of THC, CBD, CBG, and cannabinol (CBN) (a “by-product” cannabinoid resulting from the degradation of THC).

### 2.3. Data Analysis

We used IBM SPSS v22.0 software (Armork, NY, USA) for Windows (California, US). Continuous variables were shown as means with standard deviation, and categorical variables were shown as frequencies. The concentration of cannabinoids was presented according to the sampling zone in each Peruvian region. The concentration was presented as mean and standard deviation, and a 95% confidence interval (CI) was used.

## 3. Results

A total of eight samples were analyzed in 28 chromatographic assays. One sample was collected in the region of Cajamarca (F1), and three samples were from Lima (F2–F4), while Trujillo (F5–F6) and Ayacucho (F7–F8) each had two samples from their regions. The average concentrations of CBG, CBD, and THC were 1.52 ± 0.21 (CI 95%, 1.28 to 1.75), 1.20 ± 0.57 (CI 95%, 0.56 to 1.84), and 6.83 ± 3.81 (CI 95%, 4.19 to 9.47), respectively (Figure 4). The distribution of cannabis samples found in Peru is shown in Figure 2.

Table 2 shows the potential therapeutic uses of the collected samples according to the FHA 2014 monograph and the updates to it made by the author, shown in Figure 1. Most of the cannabis samples had high concentrations of THC; only one sample had less than one percent. Three samples contained CBG in low concentrations, less than 2 percent, and only two species from Lima and one from Cajamarca contained CBD, but in low concentrations. Regarding psychotropic effects, three of the samples could have large psychotropic effects, three samples could have moderate effects, one sample could have a very small effect, and one sample would have a null psychotropic effect.

## 4. Discussion

Peru is a country with an ancestral tradition of using phytotherapy, and the population normally uses medicinal plants concomitantly with their medical treatments; this could explain the rapid acceptance of the use of cannabis for medicinal therapy. Although regulation is not completely effective and the possibility of obtaining cannabis and derivatives in pharmacies is still limited, the population considers the use of this plant legitimate, and people decide to use it outside the health system. All this converges in the fact that 70% of the population of cannabis user patients continues acquiring artisanal products [35], which come mainly from national cannabis cultivars. Before the approval of the regulation of the medicinal use of cannabis in Peru, the focus of scientific studies regarding the use of cannabis was generally limited to its harms in terms of mental health, and this paradigm remains solid in many sectors; thus, studies like this could open the way to generating knowledge and interest in botanical collection, identification, analysis of chemical characteristics, and studies of the therapeutic use of a plant of high medicinal value that, outside of its legal status, is present in Peruvian territory [36]. Talking about the potential therapeutic effects of Peruvian cannabis cultivars intended for nonmedicinal use or adult use invites us to reflect on whether this use means harm in all cases; if there is an underdiagnosed population of patients who find in cannabis an improvement in their quality of life; or if the incineration of all informal/illegal crops by the authorities is efficient and ecological. Perhaps we can respond to these facts in a different way, based on new scientific findings, with more flexible cannabis control regulations for patients and more appropriate policies for the use of resources from agriculture, in a country that needs improvements in these three aspects.

Future studies with a larger number of samples, representing all 24 departments of Peru within the three main regions (coast, highlands, and jungle), and using registered genetics, are necessary to explore the relationships between this plant, its environmental conditions, and the therapeutic effects it exhibits. Additionally, there is a risk of losing information about cannabis cultivars adapted to Peruvian geography, information which could be advantageous in light of the arrival of foreign varieties during the upcoming period of national production implementation for medicinal and industrial cannabis derivatives. 

### 4.1. Strengths

This study is the first study in Peru that involved (i) collecting cannabis samples from different regions, (ii) performing an official botanical identification, and (iii) developing and performing a technique for cannabinoid quantification. Due to the increasing interest in the medicinal use of cannabis in Peru, it is now possible to study the plant from a different perspective, focusing on potential therapeutic benefits rather than health risks [21,34]. Another strength lies in the description of potential therapeutic uses of the cultivars found in different geographical regions, which could guide researchers in further investigations in this area, patients in the knowledge of the products are using, and producers in the future industrialization of cannabis or cannabinoid production for medicinal purposes, taking advantage of the adapted chemotypes that possibly develop in each region of the country.

### 4.2. Main Findings

In the four regions of Peru, there were national cannabis cultivars prior to the implementation of medicinal regulation. This regulation in theory allows the cultivation and production of cannabis derivatives; however, cannabis user patients have, until now, been supplied with imported raw materials and artisanal products from informal national cultivars [37]. Although various research laboratories in universities, industries, and police departments have liquid and gas chromatography equipment, as of the time of this study, there is no published evidence of analytical methodologies to quantify cannabinoids, making it impossible to carry out agricultural and biomedical research or quality control services.

All cannabis chemotypes found in this study have detectable concentrations of THC, and in the majority of samples this concentration is high. This could be because the informal/illegal cannabis market in Peru has historically been intended for nonmedicinal use (or adult use) in which the psychotropic effect is valued; fortunately, THC is also the molecule that has the most evidence for its usefulness in the most common medical areas among patients seeking treatment with cannabis, such as in the areas of chronic pain management, antiemetic treatment, or anti-inflammatory treatment, among others. On the other hand, three of the eight samples have detectable concentrations of CBD; one of these samples (F4) has double the concentration of CBD over THC, making it potentially useful for cases of refractory epilepsy, which is why these nonmedicinal-purpose cannabis cultivars have been appreciated by the more than 40 thousand patients registered to date in the country. However, it is necessary to complement these studies with quality controls to avoid bacterial infections, pesticide poisoning, and heavy metal poisoning. 

Sample collection was constrained to regions where limited permissibility for gathering could be obtained, due to the stigma associated with cannabis cultivation, which is often linked with illicit drug trafficking. The collection phase spanned four months, and the volume of samples and the number of regions were both relatively modest. In this study, samples were only accessible from 4 out of the 24 regions in Peru. Furthermore, this research marked the inaugural instance of the chromatographic quantification of cannabinoids in Peru. Procuring cannabinoid standards from international sources incurred significant expenses, necessitating a more extensive array of standards to yield more detailed findings. Additionally, the outcomes are presented in terms of total THC and total CBD. This was due to the limitations of the available equipment for this investigation, which employed a gas chromatograph. The injection temperature for this equipment exceeded the decarboxylation temperature of the cannabinoids, rendering it challenging to identify and quantify acidic cannabinoids. Consequently, this study focused on the overall concentrations of THC and CBD as opposed to individual cannabinoid variants.

## 5. Conclusions

Cannabis cultivars, that existed before Peruvian medicinal regulation, have potential therapeutic effects. Due to the variety in cannabinoid chemotypes found in cannabis samples from four different regions, these potential therapeutic effects can vary according to the concentrations of the cannabinoids that show major evidence of therapeutic effects, like THC, CBD, and CBG. The most important effects include the following: managing chronic pain, antiemetic, anti-inflammatory, and antipruritic effects, beneficial effects in treating duodenal ulcers, use in bronchodilators, use as a muscle relaxant, treating refractory epilepsy, anxiolytic effects, sebum reduction, effectiveness on MRSA, proapoptotic effects in breast cancer, and effectiveness in addiction and psychosis treatments, as well as effectiveness in controlling psoriasis and in treating glioblastoma. This potential invites us to continue botanical collection and select interesting chemotypes in agronomic studies with a view to proposing biomedical, observational, or clinical research.

On the other hand, having obtained concentrations of THC, we were able to propose a simple scale of the psychotropic capacity of said samples. Seven samples have psychotropic effects, and one of them fits within the legal category of “non-psychoactive cannabis”, which represents less than one percent according to Peruvian regulations. This information could be useful for patients and health professionals, since the majority of the population of Peruvians who use cannabis for medicinal purposes is supplied by the illegal/informal cannabis market. The conservation of these cultivars could be necessary in anticipation of the arrival of foreign genetics during the upcoming period of medicinal and industrial cannabis regulation implementation. Additionally, completing a catalogue of national cultivars and eventually exploring causal relationships between cannabis plant cannabinoid proportions and the environmental conditions of the region of origin are essential. It is possible and necessary to continue chemically, morphologically, and genetically identifying and quantifying cultivars from other regions of Peru, as well as developing and implementing more chromatographic techniques for cannabinoid quantification in universities, research institutes, and public and private quality control laboratories in response to the needs of patients, health care professionals, researchers, and companies dedicated to this field, as these services are currently not available.

## Figures and Tables

**Figure 1 biomedicines-12-00306-f001:**
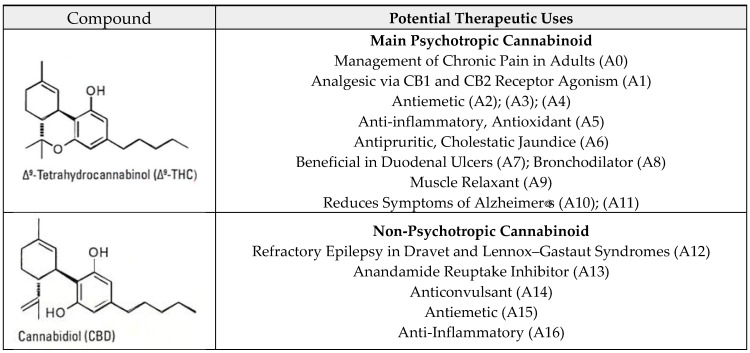
“Potential therapeutic uses of main cannabinoids” according to AHP 2014 monograph. The complete list of updated research on therapeutic potential can be found in Appendix A.

**Figure 2 biomedicines-12-00306-f002:**
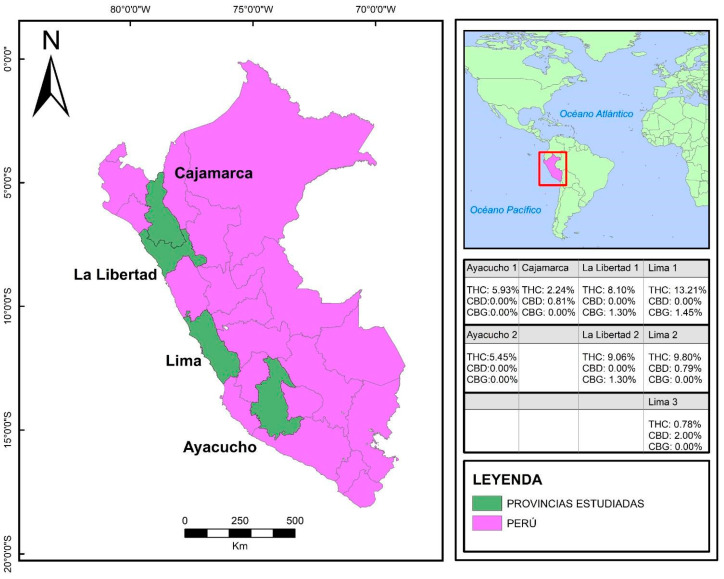
Sampling zones of cannabis cultivars in Peru. The cannabinoid concentration of each sample is shown by region in the coast and highlands. Created using ARGics v13.4. Abbreviations: THC: Δ9-tetrahydrocannabinol, CBD: cannabidiol, CBG: cannabigerol.

**Figure 3 biomedicines-12-00306-f003:**
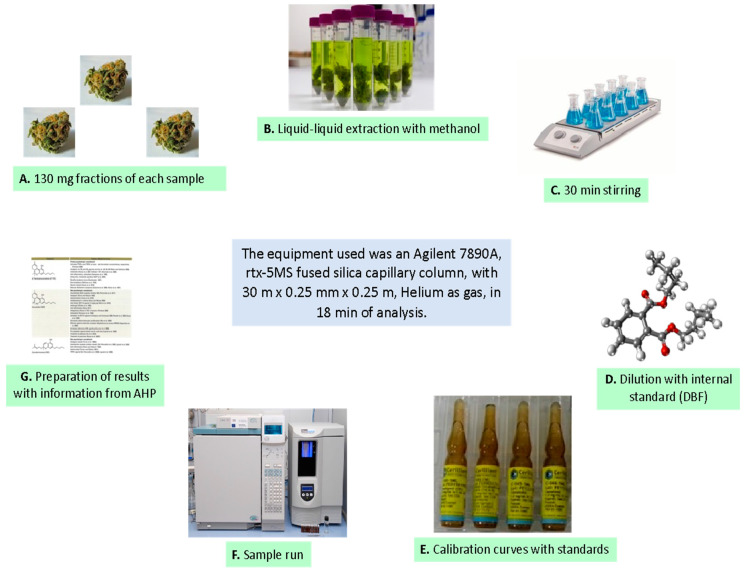
Experimental analysis for the quantification of cannabinoids in dried cannabis inflorescences obtained from cultivars in four regions of Peru.

**Figure 4 biomedicines-12-00306-f004:**
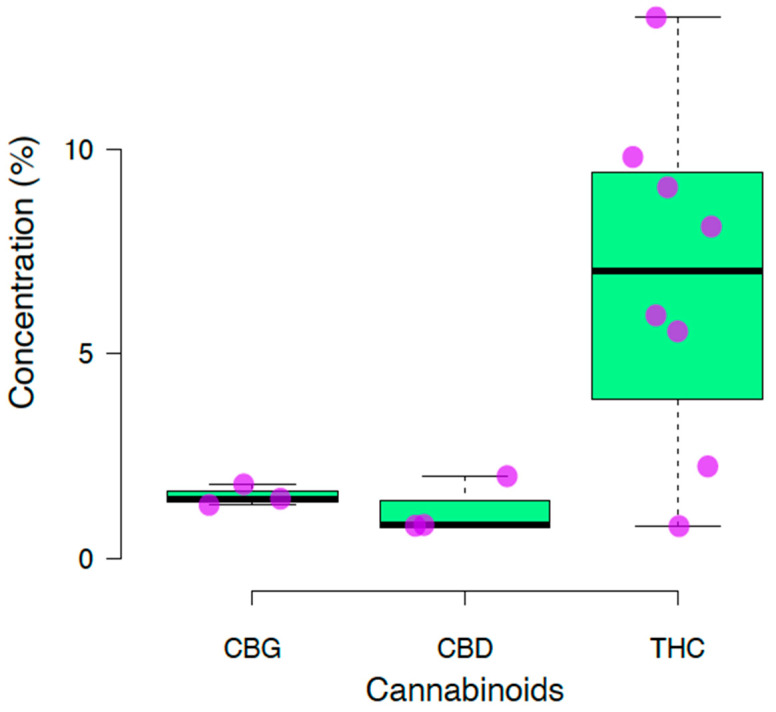
Cannabinoid profiles identified in cannabis crops in Peru. The pink dot are the samples analyzed. Abbreviations: THC: Δ9-tetrahydrocannabinol, CBD: cannabidiol, CBG: cannabigerol.

**Table 1 biomedicines-12-00306-t001:** Baseline geographic characteristics of Peruvian sampling zones.

Region	M.A.S.L.	Climate	Average Humidity (%)	Average Temperature °C/F
Lima	200	Arid subtropical	80	17.4 (63.3)
Ayacucho	2760	Inter-Andean valley	56	17.5 (63.5)
Trujillo	34	Arid subtropical	89	19.7 (67.46)
Cajamarca	2750	Inter-Andean valley	72	13 (55.4)

**Table 2 biomedicines-12-00306-t002:** Potential uses and therapeutic effects of cannabinoids according to the Peruvian geographical region. (Figure 1).

Sample	Region of Origin	Geographical Region	Cannabinoid Profile	Potential Therapeutic Effect
F1	Cajamarca	Highlands	THC: 2.24% CBD: 0.81% CBG: ND	Potency of effect: Low. Medical areas of potential: Chronic pain management, antiemetic use, anti-inflammatory use, antipruritic use, duodenal ulcer treatment, bronchodilators, muscle relaxants, and treating Alzheimer’s disease symptoms due to the presence of THC. Refractory epilepsy treatment, anxiolytic use, sebum reduction, proapoptotic use in breast cancer, and are effective on MRSA and in addiction and psychosis treatments due to the presence of CBD. Psychotropic effect: Very small.
F2	Lima	Coast	THC: 13.21% CBD: ND CBG: 1.45%	Potency of effect: Very high. Medical areas of potential: Chronic pain management, antiemetic use, anti-inflammatory use, antipruritic use, duodenal ulcer treatment, bronchodilators, muscle relaxants, and treating Alzheimer’s disease symptoms due to the presence of THC in high amounts; effective on MRSA, in controlling psoriasis, and in glioblastoma treatment due to the presence of CBG. Psychotropic effect: Large.
F3	Lima	Coast	THC: 9.80% CBD: 0.79% CBG: ND	Potency of effect: High. Medical areas of potential: Chronic pain management, antiemetic use, anti-inflammatory use, antipruritic use, duodenal ulcer treatment, bronchodilators, muscle relaxants, and treating Alzheimer’s disease symptoms due to the presence of THC. Refractory epilepsy treatment, anxiolytic use, sebum reduction, proapoptotic in breast cancer, and are effective on MRSA and in addiction and psychosis treatments due to the presence of CBD. Psychotropic effect: Moderate.
F4	Lima	Coast	THC: 0.78% CBD: 2.0% CBG: ND	Potency of effect: Moderate. Medical areas of potential: Chronic pain management, antiemetic use, anti-inflammatory use, antipruritic use, duodenal ulcer treatment, bronchodilators, muscle relaxants, and treating Alzheimer’s disease symptoms due to the presence of THC. Refractory epilepsy treatment, anxiolytic use, sebum reduction, proapoptotic in breast cancer, and are effective on MRSA and in addiction, and psychosis treatments due to the presence of CBD. Psychotropic effect: Null.
F5	La Libertad	Highlands	THC: 8.1% CBD: ND CBG: 1.30%	Potency of effect: High. Medical areas of potential: Chronic pain management, antiemetic use, anti-inflammatory use, antipruritic use, duodenal ulcer treatment, bronchodilators, muscle relaxants, and treating Alzheimer’s disease symptoms due to the presence of THC in high amounts; effective on MRSA, in controlling psoriasis, and in glioblastoma treatments due to the presence of CBG. Psychotropic effect: Large.
F6	La Libertad	Highlands	THC: 9.06% CBD: ND CBG: 1.83%	Potency of effect: High. Medical areas of potential: Chronic pain management, antiemetic use, anti-inflammatory use, antipruritic use, duodenal ulcer treatment, bronchodilators, muscle relaxants, and treating Alzheimer’s disease symptoms due to the presence of THC in high amounts; effective on MRSA, in controlling psoriasis, and in glioblastoma treatments due to the presence of CBG. Psychotropic effect: Large.
F7	Ayacucho	Highlands	THC: 5.93% CBD: ND CBG: ND	Potency of effect: Moderate. Medical areas of potential: Chronic pain management, antiemetic use, anti-inflammatory use, antipruritic use, duodenal ulcer treatment, bronchodilators, muscle relaxants, and treating Alzheimer’s disease symptoms due to the presence of THC; small psychotropic effect. Psychotropic effect: Moderate.
F8	Ayacucho	Highlands	THC: 5.45% CBD: ND CBG: ND	Potency of effect: Moderate. Medical areas of potential: Chronic pain management, antiemetic use, anti-inflammatory use, antipruritic use, duodenal ulcer treatment, bronchodilators, muscle relaxants, and treating Alzheimer’s disease symptoms due to the presence of THC; small psychotropic effect. Psychotropic effect: Moderate.

## Data Availability

Data will be disclosed upon request to the authors.

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
