# Peer review of "Therapeutic Potential of Cannabinoid Profiles Identified in Cannabis L. Crops in Peru"

_biomedicines, 2024, doi:10.3390/biomedicines12020306_

Round 1
Reviewer 1 Report
It is really interesting data with novel analysis data of cannabis. However, one request should be satisfied. As a strict regulation of cannabis, authors should get an approval number even for research purposes. Therefore, provide authorized or approval number in Peru regulation system or equivalent document(s) with legal effect for the research.
Author Response
Response to reviewers
biomedicines-2596764
REVIEWER 1
It is really interesting data with novel analysis data of cannabis. However, one request should be satisfied. As a strict regulation of cannabis, authors should get an approval number even for research purposes. Therefore, provide authorized or approval number in Peru regulation system or equivalent document(s) with legal effect for the research.
RESPONSE: Indeed, the regulation of Cannabis in Peru requires obtaining a “license for scientific research”, however this license is intended for work that requires activities such as: “importation, storage, cultivation, harvesting, propagation, transportation and manufacturing of derivatives.” .
Textual quote Article 7 of the regulation (DS 005-2019SA): “Given the diversity of scientific research projects, scientific research licenses include all activities that may be necessary to comply with the previously approved research protocol: import, storage , cultivation, harvesting, propagation, transportation and manufacturing of derivatives.”
For example, the “License for universities and agricultural research institutions” (Article 9 of the regulations), intended for the cultivation of cannabis, an important requirement is: Georeferenced location of the place where it will be carried out; or the “License for scientific research on human beings” (Article 8 of the regulations), where two important requirements are: “Directory Resolution Number that authorizes the conduct of the Clinical Trial issued by the corresponding body of the NIH”, “Copy of the authorization and/or accreditation and certification of the production laboratory in the event that the research product is manufactured in the country”, and “Copy of the Directorial Resolution that authorizes the importation of the research product, issued by DIGEMID.”
This research collects samples in a similar way to a “quality control” task, where the researchers only limit themselves to taking samples, not growing the plant and not marketing (import). Even the botanical identification of the species (carried out at the Natural History Museum) is carried out after the sample is taken. This type of descriptive research does not require: import, storage, cultivation, harvesting, propagation, transportation and manufacturing of derivatives, which is why it is not necessary to obtain a license. Finally, it is important to mention that in Peru, the consumption, possession and even cultivation of cannabis is not punishable if there is no intention to carry out illicit trafficking. So even before the 2017 law on the medicinal use of cannabis existed, these types of studies were possible within the law, it was just that there was no interest on the part of researchers.
*The main author of this research has been a technical consultant to the Congress of the Republic of Peru in the drafting of Law 30681, Law 31312 and regulation DS005-2019SA.
https://www.gob.pe/institucion/minsa/normas-legales/262787-005-2019-sa
Reviewer 2 Report
1. The title has not been reflected to the conclusion; therapeutic potential of Cannabinoids profiling is hardly addressed in the conclusion
2. In the lines 32-24; how the conclusion could be made? Does the higher levels of THC in particular region confirm the therapeutic effects...............a big flaw
3. Using the full form at least once is a prerequisite to use the abbreviation
4. Its cross-sectional study, what were the inclusion and exclusion criteria of sampling?
5. Research problem and research question is not clear in the intro section
6. Its an original research but the arrangement of the manuscript looks its a review. Cross-sectional study means an observational study. Observation on what variables should be clarified.
7. Authors have conducted GC-MS analysis? what were the array of chemotypes? Why no GCMS data is added? NO Library is referred.
8. Botanical characterization needs standardization and taxonomical identification has been disclosed using accession number
9. How methanol has been chosen for terpenophenolics? Because methanol extracts both polar and nonpolar. What were the polarity of the target chemotypes?
10 Data are not presented appropriately (1.20±0.57 wrong/ 1.20 ± 0.57 ok)
11. Discussion is not complete, results-oriented and reflexive
12. Main findings----------This part does not represent the title of the manuscript. Therapeutic potential is not in the major findings.........
13. Limitation-------its redundant to make a new section heading to limitation. Merge it with conclusion
14. Conclusion should be extensively summarized giving core impact and prospect/drawbacks/overcoming strategy
15. Line 385-87, inappropriate/unexpected
16. Line 408---------------what do you mean by this? Do you want to mean the geographical distribution of the Can-------in Peru?

please see the attachment
Author Response
Response to reviewers
biomedicines-2596764
REVIEWER 2
- The title has not been reflected to the conclusion; therapeutic potential of Cannabinoids profiling is hardly addressed in the conclusion
RESPONSE: The proposed adjustments were made.
- In the lines 32-24; how the conclusion could be made? Does the higher levels of THC in particular region confirm the therapeutic effects...............a big flaw
RESPONSE: The meaning of the statement has been corrected.
- Using the full form at least once is a prerequisite to use the abbreviation
RESPONSE: It has been corrected
- Its cross-sectional study, what were the inclusion and exclusion criteria of sampling?
RESPONSE: This information has been included in the corresponding section
- Research problem and research question is not clear in the intro section
RESPONSE: It has been corrected-
- Its an original research but the arrangement of the manuscript looks its a review. Cross-sectional study means an observational study. Observation on what variables should be clarified.
RESPONSE: The study design has been adjusted. This study is observational on the types of cultivation, the amount of cannabinoid and the therapeutic potential
- Authors have conducted GC-MS analysis? what were the array of chemotypes? Why no GCMS data is added? NO Library is referred.
RESPONSE: No, the analysis was conducted with GC-FID, because it is the only equipment available from the Faculty of Pharmacy of the Universidad Nacional Mayor de San Marcos, this detector does not work with a library, only with standards, analyzing retention times
- Botanical characterization needs standardization and taxonomical identification has been disclosed using accession number
RESPONSE: The botanical characterization was carried out at the Natural History Museum of the Universidad Nacional Mayor de San Marcos, an official institution that provides the Service of Taxonomic Study and Determination of Botanical Samples and Certification for scientific research on plants in the country subject to thesis work. undergraduate and graduate. We attach documentation.
https://museohn.unmsm.edu.pe/herbario.html
- How methanol has been chosen for terpenophenolics? Because methanol extracts both polar and nonpolar. What were the polarity of the target chemotypes?
RESPONSE: Methanol was selected as a solvent due to previous successful experiences to quantify cannabinoids through gas chromatography in methodologies adopted in the pharmacognosy laboratory of the University of La República - Uruguay.
https://www.frontiersin.org/articles/10.3389/fpls.2023.1025932/full
https://acervodigital.ufpr.br/bitstream/handle/1884/73466/R%20-%20D%20-%20FABIANO%20SOARES%20DE%20ARAUJO.pdf?sequence=1&isAllowed=y
10 Data are not presented appropriately (1.20±0.57 wrong/ 1.20 ± 0.57 ok)
RESPONSE: It has been corrected
- Discussion is not complete, results-oriented and reflexive
RESPONSE: It has been corrected
- Main findings----------This part does not represent the title of the manuscript. Therapeutic potential is not in the major findings.........
RESPONSE: It has been corrected
- Limitation-------its redundant to make a new section heading to limitation. Merge it with conclusion
RESPONSE: It has been corrected
- Conclusion should be extensively summarized giving core impact and prospect/drawbacks/overcoming strategy
RESPONSE: It has been corrected
- Line 385-87, inappropriate/unexpected
RESPONSE: It has been corrected
- Line 408---------------what do you mean by this? Do you want to mean the geographical distribution of the Can-------in Peru?
RESPONSE: Graphic was deleted
Reviewer 3 Report
Dear Authors
thank you for contributing your intersting work. You'll find the comments within the following table.
|
Section |
Page |
Line |
Comment(s) |
|
Abstract |
1 |
29 |
CBG, CBD and THC. The full-word should be given at the first time and the abbreviations within parenthesis
|
|
Introduction |
2-4 |
----- |
Although, the fragmentation of “Introduction” into sub-sections is interestingly good, the style of the journal should be strictly followed. |
|
Introduction |
2 |
69 |
The information contained includes Should be replaced with “The information within the review includes |
|
Introduction |
2 |
91-92 |
“antidrug authorities consider 91 abuse purposes lifetime prevalence is 14.8 “ This statement is not clear and confusing. Please rewrite it. Do you mean that the average age is 14.8 Y. |
|
Introduction |
4 |
138 |
“ Update of Table 6”, I think it should be deleted. Is this right?
|
|
MM |
5 |
186 |
Only 8 samples?! The number of samples is inadequate. It will be better if it more than this. |
|
MM |
6 |
198-199 (Figure 2) |
The font size is too small. It will be better if you apply larger font size to denote concentrations of cannabinoid. |
|
MM |
6 |
220-221 |
Where had, the cannabinoid quantification by a chromatography technique, been performed? I think it will be better to specify that whenever possible. |
|
MM |
7 |
223-226 (Figure 3) |
Please, apply larger font size for the labeling of images.
|
|
Results |
7 |
241-242 |
One sample 241 was collected in the region of Cajamarca (F1) ; Again, the number of samples is inadequate. It should be more than this. |
|
Results |
8 |
253-257 |
“Potential therapeutic effects….” , I think, it is not suitable to state that here in the results section. It should be in the Discussion.
|
|
Results |
8-9 |
Table 2 |
|
|
Discussion |
10-11 |
----- |
Although, the fragmentation of Discussion into sub-sections is interestingly good, the style of the journal should be strictly considered. |
|
Conclusion |
|
----- |
Good |
|
References |
|
----- |
Good and updated |
minor changes are required
Author Response
RESPONSE TO REVIEWER 3
|
Section |
Page |
Line |
Comment(s) |
|
Abstract |
1 |
29 |
CBG, CBD and THC. The full-word should be given at the first time and the abbreviations within parenthesis RESPONSE: Correction was made. |
|
Introduction |
2-4 |
----- |
Although, the fragmentation of “Introduction” into sub-sections is interestingly good, the style of the journal should be strictly followed.
RESPONSE: This section is maintained since the magazine allows division into sections. |
|
Introduction |
2 |
69 |
The information contained includes Should be replaced with “The information within the review includes RESPONSE: Correction was made. |
|
Introduction |
2 |
91-92 |
“antidrug authorities consider 91 abuse purposes lifetime prevalence is 14.8 “ This statement is not clear and confusing. Please rewrite it. Do you mean that the average age is 14.8 Y. RESPONSE: Correction was made |
|
Introduction |
4 |
138 |
“Update of Table 6”, I think it should be deleted. Is this right?
RESPONSE: Correction was made |
|
MM |
5 |
186 |
Only 8 samples?! The number of samples is inadequate. It will be better if it more than this.
RESPONSE: We accept that the samples are small, however this study managed to access 8 sampling areas in Peru for the first time. There is a difficulty in accessing crops of this type that do not have commercial or illegal purposes and therefore sampling can be valuable. |
|
MM |
6 |
198-199 (Figure 2) |
The font size is too small. It will be better if you apply larger font size to denote concentrations of cannabinoid.
RESPONSE: |
|
MM |
6 |
220-221 |
Where had, the cannabinoid quantification by a chromatography technique, been performed? I think it will be better to specify that whenever possible. |
|
MM |
7 |
223-226 (Figure 3) |
Please, apply larger font size for the labeling of images.
RESPONSE: The font size is small, however the real images have higher quality and the size of the images in the text can be larger. We ask you to review the figure files |
|
Results |
7 |
241-242 |
One sample 241 was collected in the region of Cajamarca (F1) ; Again, the number of samples is inadequate. It should be more than this. RESPONSE: We hope to continue this work by collecting more samples, but as we mentioned in section 2.2.1 Crops and sampling, it is still difficult to have access to samples of national cultivars, even more so of "local" varieties, since all crops are considered illegal. by the police and fearful citizens prefer not to have to face these procedures. |
|
Results |
8 |
253-257 |
“Potential therapeutic effects….” , I think, it is not suitable to state that here in the results section. It should be in the Discussion. RESPONSE: Correction was made |
|
Results |
8-9 |
Table 2 |
|
|
Discussion |
10-11 |
----- |
Although, the fragmentation of Discussion into sub-sections is interestingly good, the style of the journal should be strictly considered. |
|
Conclusion |
|
----- |
Good |
|
References |
|
----- |
Good and updated |
Round 2
Reviewer 2 Report
Authors have addressed all the suggestions and recommendations